# Ultra-Stable Temperature Controller-Based Laser Wavelength Locking for Improvement in WMS Methane Detection

**DOI:** 10.3390/s23115107

**Published:** 2023-05-26

**Authors:** Fupeng Wang, Jinghua Wu, Rui Liang, Qiang Wang, Yubin Wei, Yaopeng Cheng, Qian Li, Diansheng Cao, Qingsheng Xue

**Affiliations:** 1Faculty of Information Science and Engineering, Engineering Research Center of Advanced Marine Physical Instruments and Equipment, Ocean University of China, Qingdao 266100, China; jinghua@stu.ouc.edu.cn (J.W.); liangrui@stu.ouc.edu.cn (R.L.); chengyaopeng@stu.ouc.edu.cn (Y.C.); liqian@ouc.edu.cn (Q.L.); cds@ouc.edu.cn (D.C.); 2State Key Laboratory of Applied Optics, Changchun Institute of Optics, Fine Mechanics and Physics, Chinese Academy of Sciences, Changchun 130033, China; wangqiang@ciomp.ac.cn; 3Laser Institute, Qilu University of Technology (Shandong Academy of Sciences), Jinan 250102, China; wyb9806@qlu.edu.cn

**Keywords:** index terms, temperature controller, laser wavelength locking, wavelength modulation spectroscopy, signal-to-noise ratio, methane detection

## Abstract

In the wavelength modulation spectroscopy (WMS) gas detection system, the laser diode is usually stabilized at a constant temperature and driven by current injection. So, a high-precision temperature controller is indispensable in every WMS system. To eliminate wavelength drift influence and improve detection sensitivity and response speed, laser wavelength sometimes needs to be locked at the gas absorption center. In this study, we develop a temperature controller to an ultra-high stability level of 0.0005 °C, based on which a new laser wavelength locking strategy is proposed to successfully lock the laser wavelength at a CH_4_ absorption center of 1653.72 nm with a fluctuation of fewer than 19.7 MHz. For 500 ppm CH_4_ sample detection, the 1σ SNR is increased from 71.2 dB to 80.5 dB and the peak-to-peak uncertainty is improved from 1.95 ppm down to 0.17 ppm with the help of a locked laser wavelength. In addition, the wavelength-locked WMS also has the absolute advantage of fast response over a conventional wavelength-scanned WMS system.

## 1. Introduction

Laser wavelength stabilization techniques have been widely used in various advanced applications, such as quantum communication [1], optical atomic clocks [2], photonic microwave synthesizer [3], gravitational wave detection [4], relativity test [5] and cavity ringdown spectroscopy (CRDS) for sensitive molecular detection [6], where a laser source with ultra-stable wavelength/frequency and narrow linewidth is indispensable, or at least preferred. Over the last few decades, different types of semiconductor lasers have developed rapidly toward the direction of narrow linewidth, easy tunability, compactness and cost-effectiveness. Concomitantly, a series of wavelength stabilization techniques are also proposed, including the famous Pound–Drever–Hall (PDH) method [7], the saturated absorption method [8], the spectral hole burning effect [9] and so forth.

In the field of tunable diode laser absorption spectroscopy (TDLAS) gas detection, the laser wavelength stabilization technique also plays an important role in improving detection performance. We are all aware that free-running lasers are susceptible to thermal disturbance, electronic aging, and mechanical vibration [10]. As a result, the consequent wavelength fluctuation (dozens of MHz to hundreds of MHz) would affect the stability of laser output and even broaden the laser linewidth, which is prone to undermine the detection performance of TDLAS-based gas sensing systems. Therefore, active wavelength stabilization is usually applied in practice to eliminate this kind of influence. For example, the PDH method has been applied to CRDS and cavity-enhanced absorption spectroscopy (CEAS) to lock the probe laser in cavity mode [11,12]. As to photoacoustic spectroscopy [13], Dr. Wang applied the PDH method to lock a high-finesse photoacoustic cavity in 2019, realizing an ultrasensitive detection on acetylene detection. The normalized noise equivalent absorption coefficient was improved to 1.1×10−11cm−1WHz−12, which was unprecedented sensitivity among all the photoacoustic gas sensors by that moment [14]. In addition, by virtue of the PDH method, doubly resonant photoacoustic systems were proposed and continuously developed recently, and sub-ppt gas detection sensitivity was achieved with eight decades dynamic range [15,16]. However, even though the performance of the PDH method is very attractive, the optical design and electronic control for PDH are very complicated, so it is quite challenging to apply the PDH method to field applications that demand strong robustness.

In some general conditions of wavelength modulation spectroscopy (WMS) [17,18,19,20], a multi-pass long-path cell is usually used for gas detection instead of a high-finesse optical or photoacoustic cavity, where strict wavelength locking is not essential. So, some other wavelength stabilization techniques are also studied and qualified for low-precision wavelength locking. The most frequently used method is based on the monotonicity of absorption-induced odd harmonics in the WMS system. In this method, the linear region around the zero-intersection of the first or third harmonic signal is always served as an error signal to provide feedback into the laser driving current to lock the wavelength to the absorption line center [21,22]. This method was reported widely by virtue of its simplicity and compact size. Wang et al. [23] developed a photoacoustic spectroscopy (PAS) methane sensor and successfully locked the wavelength of a DFB laser at 1.65 μm within 10.6 MHz. Zhang et al. [24] reported a 2.004 μm DFB laser locked with a frequency deviation of 6 MHz over 10 min for CO_2_ measurement. Recently, Cheng et al. [25] demonstrated a DFB laser locked at 1653.72 nm with a fluctuation of less than 406 kHz by virtue of optimized PID parameters and successfully applied it for CH_4_ detection. Compared with the PDH method, the odd harmonic-based wavelength locking technique is already simplified so much. However, it still requires a software-based servo loop and additional electronic circuits to monitor and adjust the laser current in real-time.

As a matter of fact, temperature controllers are always needed in every TDLAS- or WMS-based gas detection system to make sure that the laser diode operates at a constant temperature. So, can we figure out a strategy to stabilize the laser wavelength by the temperature controller itself without adding more software or hardware to the system anymore? If so, the first step is to develop a high-precision laser temperature controller. To serve this, a series of temperature controllers have been developed and commercialized by many manufacturers, such as Thorlabs, Wavelength Electronics, and Stanford Research Systems. Integrated circuits (ICs) and modules for the applications are also widely available on the market, such as AMC7820, WTC3243, MAX8521, LTC1923 and so on. In addition, compact, low-cost laser temperature controllers are developed in the laboratory, achieving a stability of ±0.01 °C [26,27]. However, higher temperature stability is usually needed in some conditions such as, for example, when the laser wavelength is highly sensitive to temperature. In this study, we develop a temperature controller based on a commercial chip MAX1978 and successfully improve the temperature stability to an ultra-high level of 0.5 m°C. To our knowledge, this is almost the best temperature-controlling result reported so far. Considering that the wavelength response towards the temperature of the CH_4_ laser diode used in our lab is 100 pm/°C [25], the laser wavelength can be potentially stabilized at 1653.72 nm with a fluctuation of 5.5 MHz (0.05 pm), ideally. Therefore, the second part of this study is proposing a strategy to introduce a feedback signal to the temperature controller to stabilize the laser wavelength dynamically. In the verification experiment, the central wavelength of the laser diode is locked to the CH_4_ absorption line at 1653.72 nm with a fluctuation of less than 19.7 MHz (~0.18 pm). The result reaches the same level as reported in a previous study [23], but without adding an additional software or hardware component. To our knowledge, this is the first report using this kind of strategy for laser wavelength stabilization. In the end, the temperature controller-based wavelength locking module is applied to a WMS CH_4_ sensing system to improve its detection performance and compare with conventional wavelength scanned WMS mode. For the same concentration of 500 ppm CH_4_ sample detection, the 1σ SNR is increased from 71.2 dB to 80.5 dB and the peak-to-peak uncertainty is improved from 1.95 ppm down to 0.17 ppm. In addition, the wavelength-locked WMS also has the absolute advantage of fast response speed over a conventional wavelength-scanned WMS system.

## 2. Methodology Demonstration

Considering that a temperature controller is utilized for laser wavelength locking in this study, in the very beginning, the principle of temperature controlling for the laser diode is introduced briefly based on Figure 1. For commercial applications, a negative temperature coefficient (NTC) thermistor and a thermoelectric cooler (TEC) are usually packaged together with the laser chip within the same copper shell. The resistance of the thermistor indicates the internal temperature of the laser diode, and the TEC can be controlled to heat and cool the laser chip. The relationship between the thermistor resistance $R_{t}$ and its temperature T can be expressed as:(1)Rt=R0×expB×1T−1T0
where $R_{0}$ is 10 kΩ and $T_{0}$ is 25 °C (297.15 K), B is the material coefficient which is 3950 for our laser diode used in this study. Obviously, the relationship between $R_{t}$ and T is nonlinear. In a temperature-controlling loop, the thermistor is usually connected to an H-bridge circuit as shown in Figure 1c, and the variable resistor $R_{\mathrm{set}}$ is used to set the target temperature. Differential voltage between $V_{\mathrm{Rt}}$ and $V_{\mathrm{set}}$ is calculated as error feedback to a PID controller, and then the output of the PID controller is used to modulate the TEC for temperature stabilization. At last, the H-bridge is balanced, which means the temperature is stabilized at the target. At this moment, $V_{\mathrm{Rt}}$ and $V_{\mathrm{set}}$ can be expressed:(2)$$V_{Rt}=\frac{R_{t}}{R_{1}+R_{t}}\cdot V_{ref}$$
(3)$$V_{set}=\frac{R_{set}}{R_{2}+R_{set}}\cdot V_{ref}$$

It is obvious that the relationship between $V_{\mathrm{Rt}}$ and $R_{t}$ is nonlinear as well. However, if we combine Equations (1) and (2) together and simulate the relationship between $V_{\mathrm{Rt}}$ and T in a small range, approximate negative linearity is observed, as shown in Figure 1b.

In the next part, the CH_4_ absorption line at 1653.72 nm is chosen as an example to explain how wavelength locking works. Standard 2nd and 3rd harmonic curves are simulated in Figure 2, and the maximum value $H_{2\max}$ of 2nd harmonic signal is always measured to predict gas concentration. However, the symmetric point $H_{3\mathrm{ref}}$ of 3rd harmonic signal as labeled in Figure 2 has nothing to do with the gas concentration and laser power fluctuations, which has been employed as a reference to locate the absorption center $\lambda_{0}$ for wavelength locking in many studies [21,22,23,24,25]. The monotonic region near $H_{3\mathrm{ref}}$ is utilized as the input of PID controller to dynamically stabilize the laser wavelength at $\lambda_{0}$. As to the reason why 1st harmonic signal is not selected as the reference, it is because a strong background exists in the 1st harmonic signal due to the residual amplitude modulation (RAM) effect of the DFB laser [23,28].

Comparing Figure 1a,b and the monotonicity around $H_{3\mathrm{ref}}$ in Figure 2, we find out that the 3rd harmonic signal can be connected to $V_{\mathrm{set}}$ point for automatic wavelength locking. The logic chart is depicted in Figure 3. We suppose that the laser wavelength is fixed at $\lambda_{0}$ in the very beginning by constant temperature and driving current. Disturbed by environmental factors, the laser wavelength may drift to the direction of $\lambda >\lambda_{0}$, and then the measured 3rd harmonic value would increase from the reference point $H_{3\mathrm{ref}}$, which means the $V_{\mathrm{set}}$ increase as well. Because the PID controller keeps operating all the time to make the H-bridge balanced, as a result, the thermistor voltage $V_{\mathrm{Rt}}$ also increases. The increase in $V_{\mathrm{Rt}}$ implies that the resistance $R_{t}$ of the thermistor increases, which means the laser temperature decreases referencing Figure 1b. As a result, the laser wavelength would decrease back to original $\lambda_{0}$ due to the decreased operating temperature. Similarly, if the laser wavelength drifts to the direction of $\lambda <\lambda_{0}$, the negative feedback controlling loop also works as displayed in Figure 3. Therefore, the flow chart in Figure 3 indicates that the temperature controller itself can be used for wavelength locking function so long as it links the real-time measured 3rd harmonic signal to the $V_{\mathrm{set}}$ point instead of the variable resistor $R_{\mathrm{set}}$.

## 3. Experiment Verification

### 3.1. Ultra-Stable Temperature Controller Development

In this section, we design an ultra-stable temperature controller and improve it to a higher stability level of 0.5 m°C by optimizing the PID parameters and power supply noise. Figure 4 provides the primary electrical connection of the developed temperature controller, and the H-bridge circuit mentioned in Figure 1c is connected to the differential input of the PID controller by two voltage followers. The difference between $V_{\mathrm{Rt}}$ and $V_{\mathrm{set}}$ is calculated and amplified by 50 times as the error input of the PID network ($C_{1},C_{2},C_{3},R_{3},R_{4},{\;R}_{5}$). The PID controller is a very important part of MAX1978, and its parameters ($C_{1},C_{2},C_{3},R_{3},R_{4},{\;R}_{5}$) could be configured independently outside the commercial chip. The output of the PID controller $V_{\mathrm{out}}$ is used to drive the PWM generator. The generated PWM signal is used to open and close the MOS gates to control the driving current of TEC for heating and cooling the laser diode. A single-pole double-throw (SPDT) switch is set in the diagram, and when the switch S is turned to position a, the circuit acts as a standard temperature controller. If the switch S is turned to position b, the measured 3rd harmonic signal can be introduced to the temperature-controlling loop for dynamic wavelength locking. A physical picture of the developed temperature controller is displayed in Figure 5, and pins of the laser diode and power supply can be connected to the circuit board by sockets on the top left. The reference voltage $V_{\mathrm{ref}}$ is 2.048 V, provided by the ADR420 chip, which has an ultralow noise of 1.75 μV. A variable resistor is soldered to the bottom right corner of the circuit board for $V_{\mathrm{set}}$ adjustment. The voltage $V_{\mathrm{Rt}}$ of the thermistor is monitored through an SMA interface by a multimeter (DMM6500, Tektronix, Beaverton, OR, USA) to calculate the laser diode internal temperature based on Equations (1) and (2) to evaluate the temperature stability. Another SMA interface soldered on the right is utilized for 3rd harmonic signal input. The function of the SPDT switch marked with a yellow arrow has been demonstrated in Figure 4. The circuit parameters used in this module are listed in Table 1. R_1_ and R_2_ are set to 10 kΩ to match the thermistor Rt as part of the H-bridge sampling circuit. R_set_ is a high-precision rheostat with a low-temperature drift of 50 ppm/°C to guarantee the precision of V_set_. $C_{1},C_{2},C_{3},R_{3},R_{4},{\;R}_{5}$ are key parameters of the PID controller, which are chosen based on our experience reported in a previous study [25]. $L_{1},L_{2},C_{4},R_{5}$ constitute the basic LC filter to rectify the TEC driving current from MOS gates.

In the following step, the performance of the temperature controller is tested in advance when it operates in the standard mode (the SPDT switch is turned to position a in Figure 4). A butterfly-packaged DFB laser [25] is chosen as the object for temperature control. In the very beginning, as shown in Figure 6a,b, the temperature controller is not activated, the voltage of the thermistor inside the DFB laser is recorded for a while and the corresponding temperature is calculated. It is easy to find that the internal temperature of the DFB laser changes a lot with room temperature.

Next, the temperature controller is activated, and the internal temperature of DFB laser is stabilized at the setpoint immediately. Similarly, the thermistor voltage $V_{\mathrm{Rt}}$ is monitored for nearly ten minutes, as shown in Figure 6c, and a fluctuation of 11 μV is achieved. Based on Equations (1) and (2), the temperature stability is calculated and plotted in Figure 6, and histogram analysis is performed on the database as displayed in Figure 7. As a result, we can determine that the collected temperature data has a 98% probability of falling within the 0.5 m°C range. Referencing the wavelength response of 100 pm/°C reported in our previous paper [29], this gives us confidence that it has the potential to stabilize laser wavelength within 0.05 pm (5.5 MHz) by virtue of the developed temperature controller.

In addition to the temperature stability, the time response of the temperature controller is tested as well. In the experiment, we continuously set several temperature targets by manually adjusting the variable resistor $R_{\mathrm{set}}$. We can find that the internal temperature of the DFB laser will be stabilized quickly despite the temperature being disturbed greatly. For instance, in Figure 8 inset, it takes only 9 s for the temperature controller to stabilize the DFB laser from ~36 °C to ~30 °C. In practice, the wavelength drift of the DFB laser is gradual and tiny when it operates in free-running mode. Therefore, a minor adjustment of laser temperature is able to calibrate such wavelength drift. In other words, the response speed revealed in Figure 8 is high enough to realize efficient wavelength locking to our experience.

### 3.2. Laser Wavelength Locking Evaluation

In this section, a fixed wavelength modulation spectroscopy (FWMS) [25] system is constructed as depicted in Figure 9 to verify the temperature controller for wavelength locking. A DFB laser operating at 1653 nm is chosen as the light source, which is utilized to detect the CH_4_ absorption line at 1653.72 nm. The operating characteristics of a DFB laser have been measured in our previous study [29]. Line selection for CH_4_ detection has been mentioned in references as well [25,29]. A 4 kHz sinusoidal signal (p-p 800 mV) is converted into the current to modulate the laser output by a commercial driver (LDC501, Stanford Research Systems, Sunnyvale, CA, USA) with a 25 mA/V conversion ratio; meanwhile, a 40 mA bias current is added by LDC501. Thus, the DFB laser works in the FWMS mode. The laser output is split by a 1×2 coupler into two beams, one of which is detected by a photodetector (integrated with a miniatured reference cell including constant concentration CH_4_). A transimpedance amplifier (TIA) is made to convert the photocurrent into electrical voltage, and a lock-in amplifier (LIA) is used to measure the absorption-induced harmonic signals. The 3rd harmonic signal from LIA can be connected to the developed temperature controller for wavelength locking. Another beam of 1×2 coupler is connected to a 3 m path-length gas cell for sample CH_4_ detection. In the experiment of Section 3.2, sample CH_4_ in different concentrations is provided by a gas mixing system as shown on the right side of Figure 9, including two flowmeters, pure nitrogen, and 1% methane.

In the first step of the experiment, the operating parameters of the DFB laser should be determined so that its wavelength can locate at the CH_4_ absorption center of 1653.72 nm. The driving current remains unchanged as mentioned in the above paragraph; meanwhile, the temperature is increased from 28 °C to 33 °C to realize the wavelength scanning. In the period, 2nd and 3rd harmonic values are measured by LIA and plotted in Figure 10. It is observed that when the DFB laser operates at 30.689 °C, its wavelength is located at the CH_4_ absorption center. The 2nd harmonic reaches its maximum value and the reference point $H_{3\mathrm{ref}}$ mentioned in Figure 2 is measured to be −0.07 mV. Obviously, the measured 3rd harmonic signal cannot be directly connected to the H-bridge circuit for $V_{\mathrm{set}}$ setting. Based on Equations (1)–(3), when the H-bridge circuit is balanced, the $V_{\mathrm{Rt}}$ and $V_{\mathrm{set}}$ must be 897.64 mV to make the DFB laser operate at 30.689 °C. So, the real-time measured 3rd harmonic signal should be adjusted by:(4)$$H_{3a}=H_{3r}+\left[897.64\;\mathrm{mV}-\underset{H_{3ref}}{\underbrace{\left(-0.07\;\mathrm{mV}\right)}}\right]$$
where $H_{3a}$ is the post-adjusted 3rd harmonic signal which can be connected to the H-bridge circuit already, $H_{3r}$ is the real-time measured 3rd harmonic signal by LIA.

In the experiment of wavelength locking, the thermistor voltage $V_{\mathrm{Rt}}$ is monitored to analyze the temperature fluctuation of the DFB laser as shown in Figure 11. In the very beginning, the DFB laser operates at 30.689 °C by virtue of the temperature controller working in the standard mode, which means the $V_{\mathrm{set}}$ is set to 897.64 mV by the variable resistor $R_{\mathrm{set}}$. Suddenly, the SPDT switch is turned from position a to position b (Figure 4) to start the wavelength locking mode; subsequently, the thermistor voltage jitters a lot but soon is stabilized within 4.8 s as depicted in Figure 11. Unfortunately, the voltage $V_{\mathrm{Rt}}$ stability is worse compared with the standard mode in the very beginning. There are two reasons in our opinion. One is the noise level of the measured 3rd harmonic signal is higher than the voltage $V_{\mathrm{set}}$ of $R_{\mathrm{set}}$. The noise level connected to the H-bridge circuit may be the primary factor that limits the wavelength-locking stability. Another reason may be that 3rd harmonic detection consumes time, which extends the response time of the temperature controller so that the temperature jitter cannot be corrected in time. In order to analyze the wavelength-locking stability in detail, $V_{\mathrm{Rt}}$ data marked with a dashed box in Figure 11 is plotted in Figure 12a. The corresponding temperature is calculated based on Equations (1) and (2) and plotted in Figure 12b; histogram analysis is performed as well to evaluate the temperature stability as shown in Figure 12c. The long-term recorded temperature values have a 98% probability of falling within 1.8 m°C. Referencing the wavelength response of 100 pm/°C, the laser wavelength is stabilized within 0.18 pm (19.7 MHz) by the developed temperature controller.

### 3.3. CH4 Detection Improvement Based on the Wavelength-Locked WMS System

The laser wavelength locking technique would bring a series of advantages compared to conventional a wavelength-scanned WMS system. Due to the laser wavelength being locked at the CH_4_ absorption center, the absorption-induced 2nd harmonic signal is a DC output instead of a harmonic curve. This kind of characteristic enables us to compress the bandwidth of the lock-in amplifier to suppress noise, and a cumulative averaging algorithm could be applied as well thanks to the higher response speed. Therefore, we will evaluate the SNR improvement in detail in this section. First, the conventional wavelength-scanned WMS system is constructed by adjusting the schematic in Figure 9, sawtooth waves in frequency of 10 Hz and 4 kHz sinusoidal signal are added together to drive the DFB laser. The sensing beam of the laser output is connected to a 3 m path-length gas cell for CH_4_ detection. The transmitted light is measured by a photodetector and amplified by a TIA; afterward, an LIA is employed for 2nd harmonic detection. In the experiment, the amplitude of 2nd harmonic signal is used to infer the CH_4_ concentration, and the standard deviation of a non-absorption baseline is calculated as the 1σ noise level, and then the SNR could be computed subsequently. As shown in Figure 13a, absorption-induced 2nd harmonic curves in different lock-in bandwidths from 30 Hz to 100 Hz are plotted for comparison. Obviously, the 2nd harmonic amplitude decreases with bandwidth compression. To explore the optimal bandwidth, the noise level and SNR are provided in Figure 13b. The system achieves the best SNR of 71.2 dB when the lock-in bandwidth is set to 60 Hz.

Based on the optimized 60 Hz lock-in bandwidth, several concentrations of 10 ppm, 20 ppm, 50 ppm, 100 ppm, 500 ppm, 1000 ppm and 2000 ppm are provided by the gas mixing system presented in Figure 9 and used to do the linear test. As a result, an R-square of 0.99948 is achieved as shown in Figure 14a. When a 500 ppm sample is measured for 20 s, the measurement uncertainty is estimated to be 1.95 ppm, as displayed in Figure 14b.

The results provided by the wavelength-scanned WMS system act as the control group. Afterward, the laser wavelength is locked at 1653.72 nm, as demonstrated in Section 4, and the other experimental conditions are consistent with the control group. In the very beginning, the same optimization experiment is conducted to determine the optimal lock-in bandwidth. Because the output of LIA is a DC signal in the wavelength-locked mode, the 2nd harmonic amplitude is almost unaffected by the bandwidth as shown in Figure 15. However, the noise level is continuously decreasing. At last, the wavelength-locked WMS system achieves a better SNR of 80.5 dB when the low-pass filter bandwidth of LIA is set to 5 Hz. This bandwidth is much lower than the previous optimal value of 60 Hz in a wavelength-scanned WMS system.

Based on the 5 Hz lock-in bandwidth, the same sets of CH_4_ concentrations are provided to do the linear test. As a result, an R-square of 0.99973 is achieved as shown in Figure 16a, which is a little better than wavelength-scanned WMS mode. Correspondingly, a 500 ppm sample is measured for 20 s for transient uncertainty analysis. In the conventional wavelength scanned mode, only ten valid values can be calculated and collected per second because the scanning frequency is 10 Hz. However, every data point can be used as a valid value when the laser wavelength is dynamically locked at the CH_4_ absorption center. Figure 16b displays all measured data points in 20 s at a sampling rate of 26.79 kHz. The peak-to-peak uncertainty is estimated to be 0.33 ppm, which is better than the wavelength-scanned WMS of 1.95 ppm. This improvement is attributed to the compression of lock-in bandwidth from 60 Hz down to 5 Hz. In addition, the wavelength-locked WMS mode provides a large amount of data throughput which only depends on the sampling rate of an analog-to-digital converter (ADC). This allows us to apply an averaging algorithm to further reduce the random noise. As shown in Figure 16c, 2679 times averaging is performed on the raw data in Figure 16b, giving the same response speed of 10 Hz with the wavelength scanned WMS mode. As a result, the measuring uncertainty has been further reduced to 0.17 ppm, which is improved by more than one order of magnitude compared to wavelength-scanned WMS mode, because we discuss the uncertainty improvement based on a 20 s collected dataset, which is a very short time. In this period, uncertainty coming from the circuit drift and gas mixing system could be ignored. So, the measurement uncertainty improvement between Figure 14 and Figure 16 is mainly due to the compressed lock-in bandwidth and averaging algorithm. To our knowledge, 0.17 ppm at a 10 Hz data rate is a good result for CH_4_ detection at a wavelength of 1653.72 nm. In other studies, it often takes tens to hundreds of seconds of integration time to achieve sub-ppm measurement accuracy or it needs to select a stronger absorption line at mid-infrared region [30,31,32].

## 4. Conclusions

There are several exciting things reported in this study, as follows. The first one is the ultra-stable temperature controller is achieved with a 0.0005 °C stability. During the process of development, we conclude that the parameters of the PID network and the noise level of reference voltage are decisive for making an excellent temperature controller. Considering that a temperature controller is widely required in laser spectroscopy systems, we believe this would be interesting to researchers in this field. The second one is that a new wavelength-locking strategy is proposed based on the developed temperature controller. This strategy takes advantage of the temperature controller itself without adding additional software programs or hardware circuits. Compared with the previous methods, it simplifies the system’s complexity. It should be noted that higher-precision detection of 3rd harmonic signal would further improve the performance of wavelength locking. The third one is that the temperature controller-based wavelength locking strategy is successfully applied to the WMS CH_4_ sensing system for verification. By virtue of compressed lock-in bandwidth and averaging algorithm, the system SNR is increased by 9.3 dB and measuring uncertainty is improved by one order of magnitude. We hope every part reported in this paper, the ultra-stable temperature controller, new wavelength locking strategy, and improvement of the WMS system would be helpful for other researchers who are interested in TDLAS gas sensing.

Honestly, the precision of such a temperature controller-based wavelength locking technique is not as good as the PDH method or current tuning method. It may be not competent for applications where a high-finesse optical cavity is required to stabilize. There are two approaches to further improve the wavelength locking performance. One is to further boost the temperature-controlling stability, which is very difficult to our knowledge because 0.0005 °C is already challenging. Another approach is to select laser sources whose wavelength is not sensitive to temperature fluctuation, such as, for example, the vertical cavity surface emitting laser (VCSEL) instead of the DFB laser.

## Figures and Tables

**Figure 1 sensors-23-05107-f001:**
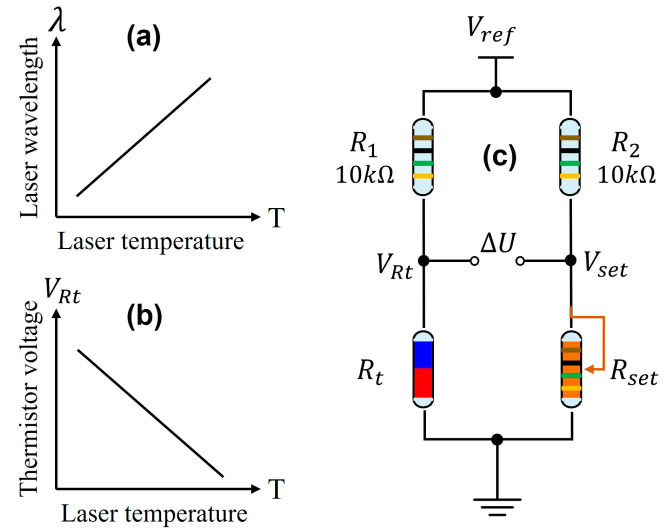
(**a**) Laser wavelength response towards operating temperature (100 pm/°C) [23]; (**b**) the relationship between NTC thermistor voltage and laser temperature; (**c**) an example of H-bridge circuit used in temperature-controlling PID loop.

**Figure 2 sensors-23-05107-f002:**
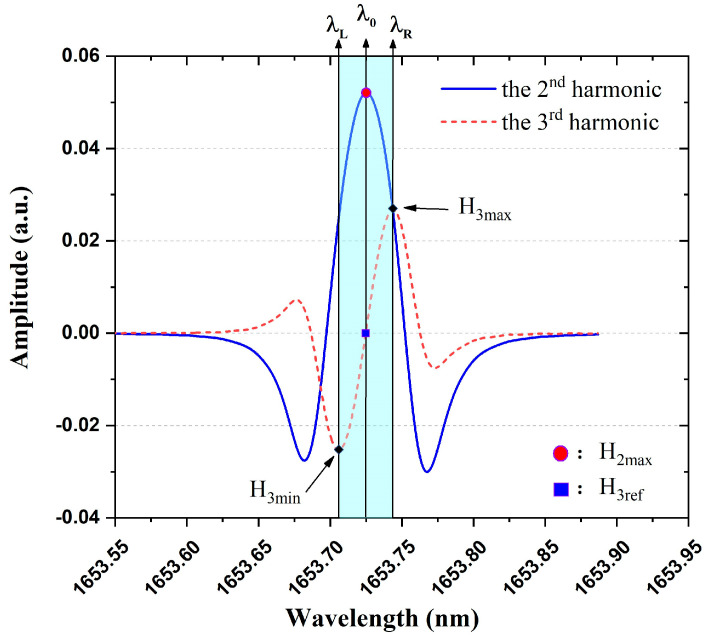
Simulated 2nd and 3rd harmonic signals based on CH_4_ absorption line at 1653.72 nm. $\lambda_{0}$ is the absorption center of CH_4_ gas.

**Figure 3 sensors-23-05107-f003:**
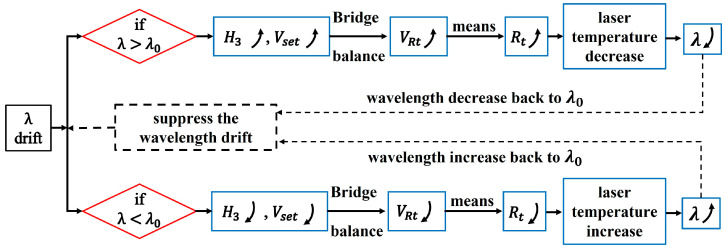
The logic chart of using temperature controller for laser wavelength locking based on the 3rd harmonic strategy.

**Figure 4 sensors-23-05107-f004:**
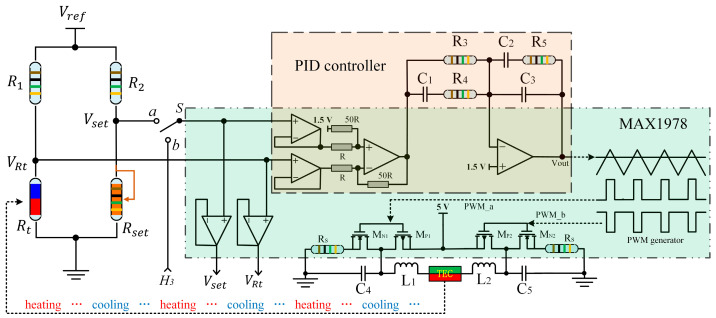
Schematic diagram of the temperature controller, $R_{t}$ and TEC are packaged inside the DFB laser copper shell. The units in the green shadow area are packaged inside the commercial MAX1978 chip. The units in the yellow shadow area constitute the PID controller, which is the key to the temperature controller.

**Figure 5 sensors-23-05107-f005:**
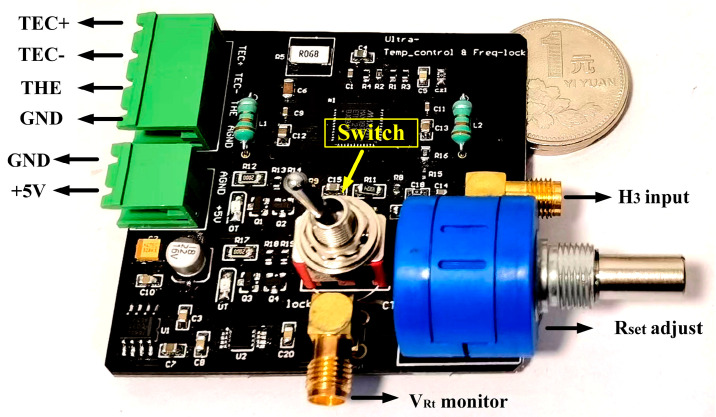
Picture of the temperature controller module. The TEC and thermistor of laser diode can be connected to TEC+, TEC-, THE and GND. A positive 5 V DC power supply is needed for this module.

**Figure 6 sensors-23-05107-f006:**
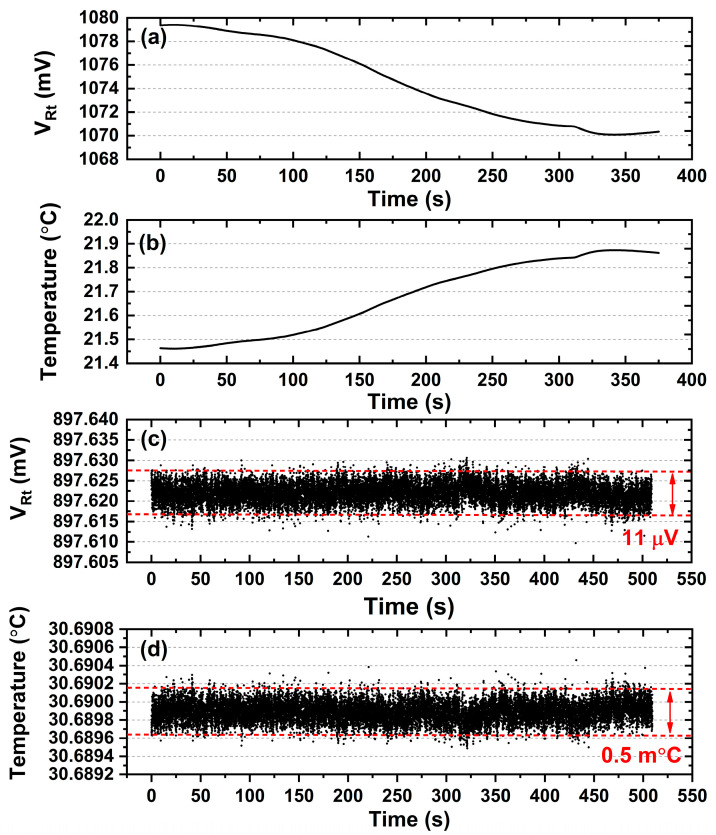
Stability test of temperature controller. (**a**,**b**) The thermistor voltage and laser diode internal temperature recorded in room atmosphere without turning on the temperature controller. (**c**,**d**) The thermistor voltage and laser diode internal temperature recorded when turning on the temperature controller.

**Figure 7 sensors-23-05107-f007:**
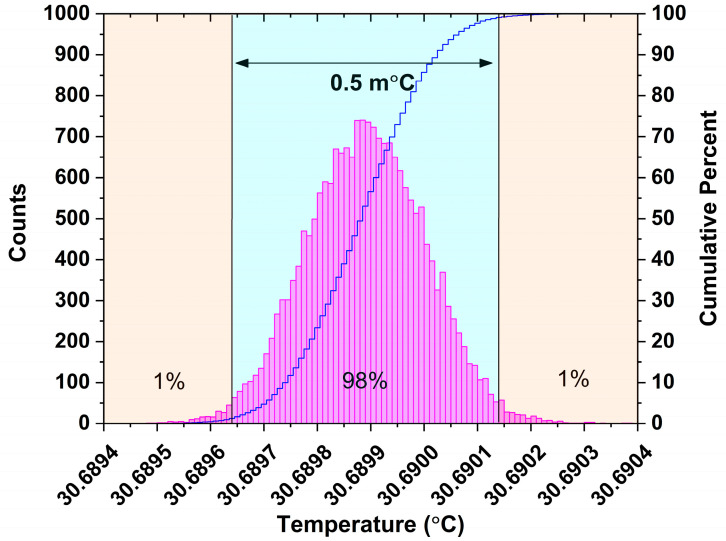
Histogram analysis of temperature stability.

**Figure 8 sensors-23-05107-f008:**
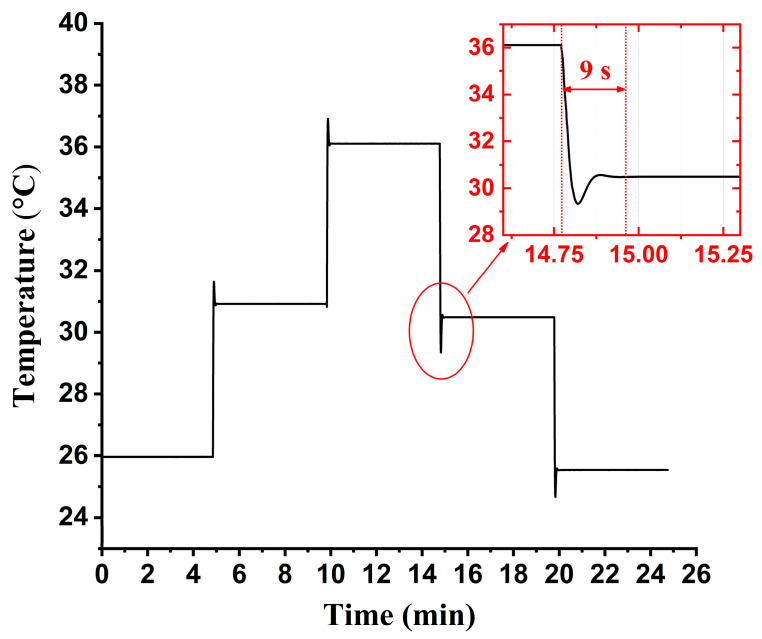
Time response of the developed temperature controller.

**Figure 9 sensors-23-05107-f009:**
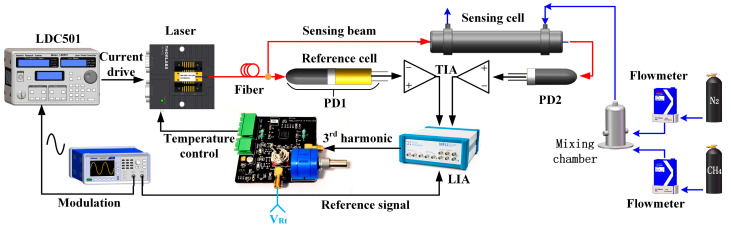
Schematic of verifying the temperature controller for wavelength locking. PD: photodetector; TIA: transimpedance amplifier; LIA: lock-in amplifier.

**Figure 10 sensors-23-05107-f010:**
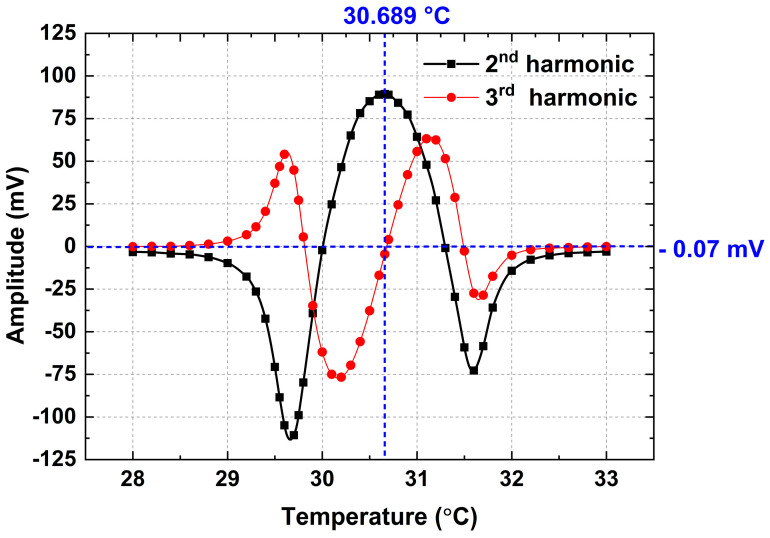
Measured 2nd and 3rd harmonic signals by modulating the temperature of DFB laser.

**Figure 11 sensors-23-05107-f011:**
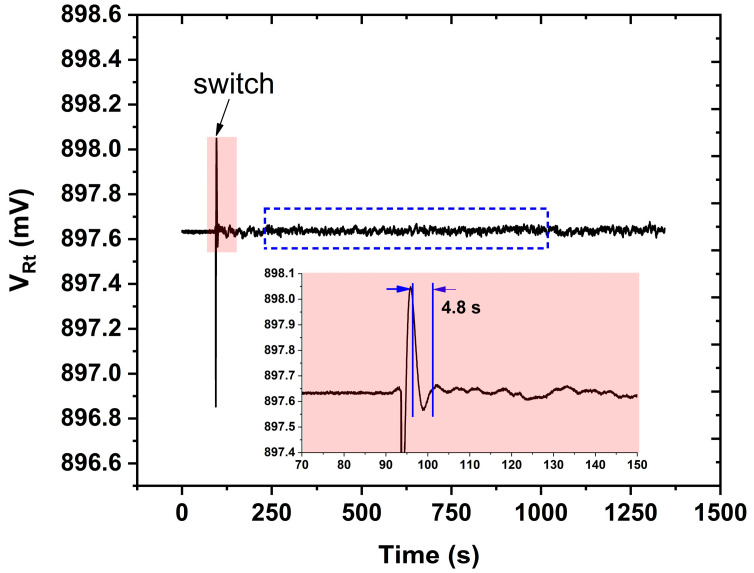
The thermistor voltage when the temperature controller switches from standard mode to wavelength locking mode.

**Figure 12 sensors-23-05107-f012:**
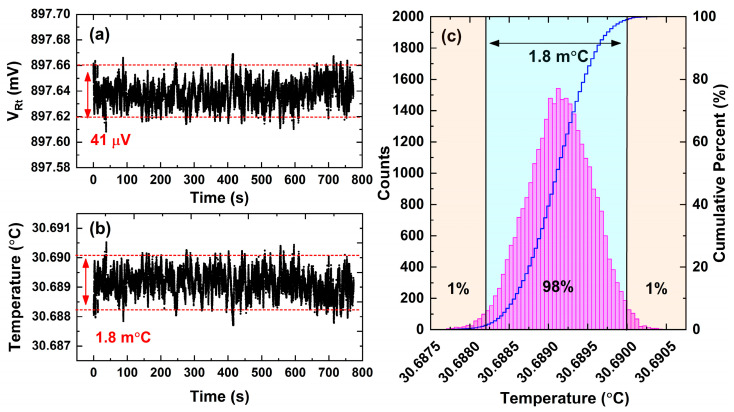
The wavelength locking analysis based on the temperature stability. (**a**) The thermistor voltage recorded when running the wavelength locking function. (**b**) Laser diode internal temperature recorded when running the wavelength locking function. (**c**) Histogram analysis of temperature stability.

**Figure 13 sensors-23-05107-f013:**
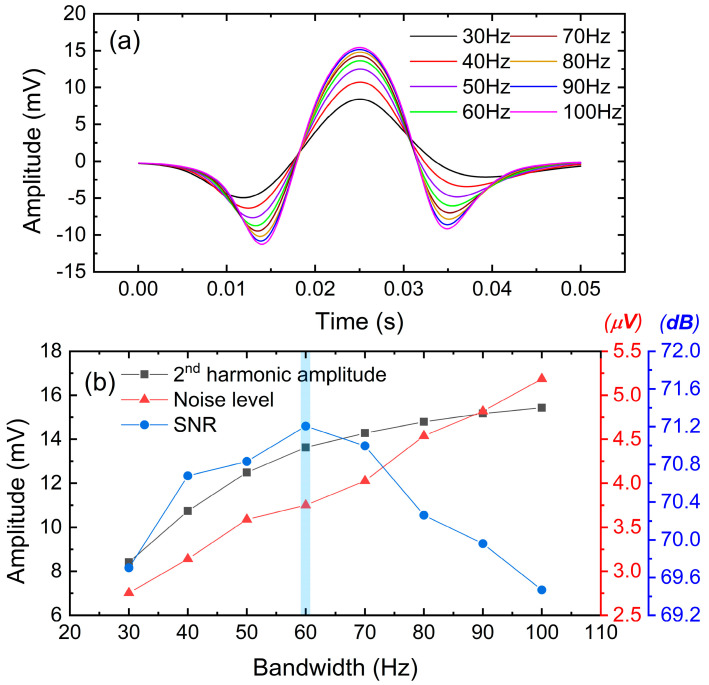
Bandwidth optimization for conventional wavelength scanned WMS system based on 500 ppm sample CH_4_; (**a**) 2nd harmonic signals measured in different lock-in bandwidths; (**b**) SNR analysis in different lock-in bandwidths.

**Figure 14 sensors-23-05107-f014:**
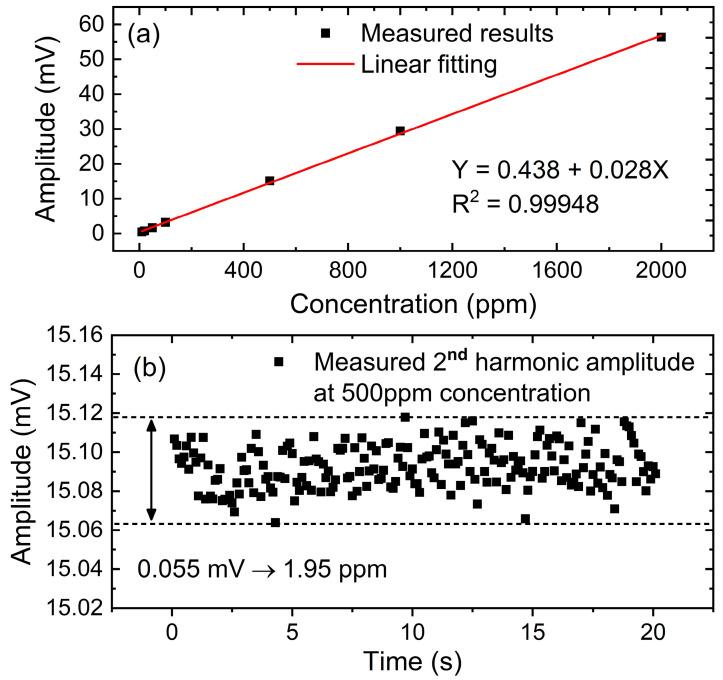
Concentration detection based on the optimized 60 Hz lock-in bandwidth. (**a**) Linear fitting of measured results in different concentrations, (**b**) Measuring uncertainty analysis based on 500 ppm sample CH_4_.

**Figure 15 sensors-23-05107-f015:**
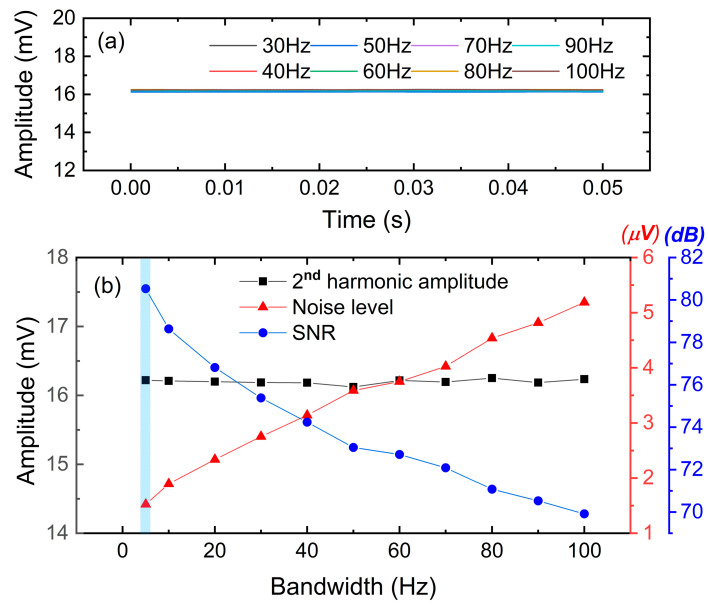
Bandwidth optimization for wavelength-locked WMS system based on 500 ppm sample CH_4_. (**a**) Second harmonic signals measured in different lock-in bandwidths. (**b**) SNR analysis in different lock-in bandwidths.

**Figure 16 sensors-23-05107-f016:**
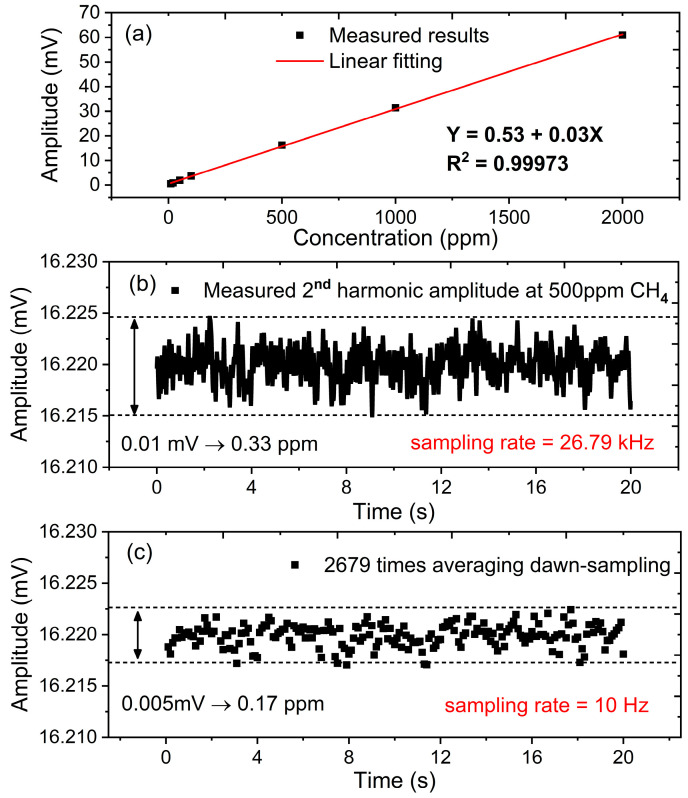
Concentration detection based on the optimized 5 Hz lock-in bandwidth. (**a**) Linear fitting of measured results in different concentrations, (**b**) Measuring uncertainty analysis based on raw data in 26.79 kHz sampling rate, (**c**) Measuring uncertainty analysis after 2679 times averaging.

**Table 1 sensors-23-05107-t001:** The important parameters used in the temperature-controlling circuit.

Designator	Value	Designator	Value
R_1_	10 kΩ	C_1_	0.47 μF
R_2_	10 kΩ	C_2_	10 μF
R_3_	1.1 MΩ	C_3_	0.047 μF
R_4_	20 kΩ	C_4_	1 μF
R_5_	75 kΩ	C_5_	1 μF
R_set_	100 kΩ variable	L_1_	3.3 μH
R_t_	10 kΩ @ 25 °C	L_2_	3.3 μH

## Data Availability

The data is available from the corresponding author upon reasonable request.

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
