# Peer review of "Ultra-Stable Temperature Controller-Based Laser Wavelength Locking for Improvement in WMS Methane Detection"

_sensors, 2023, doi:10.3390/s23115107_

Round 1

Reviewer 1 Report

See attached file for comments.

English is OK.

Author Response

Dear Reviewer,

Thank you very much for your efforts on our manuscript. The comments are all valuable and very helpful for revising and improving our study. All the comments are studied carefully and made corrections which we hope to meet with your approval. All the modifications are marked in red in the revised manuscript. The responses to your comments are reported below:

“This paper presented the development of a temperature controller with a good stability. Based on the laser wavelength locking strategy, the developed system is applied to measure methane. The SNR was examined for the methane sample detection. Overall, the paper is well organised. The fundamentals and design of the temperature controller were given and validated against experiment. The aim of the paper is suitable for potential publication on the Sensors. However, some defects have been found in the paper, which need to be fully addressed.”

  1. When reviewing general conditions of WMS in the introduction, only [17], the own work by the authors was cited. There are several recent and key literatures ([1-3] in supplementary references below) reviewing the WMS techniques and missed by the authors.

Response: Thank you for the recommendation. The three review papers are very comprehensive and instructive for us and would be interested to other readers. We have supplemented the references in the revised manuscript.

  1. The authors noted the necessity of development of the temperature controller for laser diode. It is well known that the temperature controllers are commercially available from many manufactures, such as Thorlabs and Wavelength Electronics. Why these controllers cannot satisfy the authors’ requirement in this work? Maybe this is needed to control the laser diode in the authors lab with 100 pm/degree? With another laser diode having lowering temperature drift, such a control precision may not be needed at all. In addition, there are also controllers reported with high control precision ([4-6]), but not addressed by the authors.

Response: Thank you very much for the comment. We agree with you that there are a series of temperature controllers commercially available on the market, and there are some chips specially designed for temperature controlling circuits. Actually, we also have a commercial instrument LDC501 (Stanford Research Systems, USA) in our lab, however, it is not flexible enough to support the study in this paper because the commercial instrument is highly integrated so that we can not introduce the feedback signal to the interior of the instrument.

What we need to declare is the laser wavelength response is not the determining factor why we decide to develop temperature controller ourself. To our experience, the wavelength-temperature response of DFB laser is 0.1 nm/degree ~ 0.2 nm/degree, obviously, high-precision temperature controller is very necessary in such DFB laser based systems. For VCSEL lasers, the temperature tuning coefficient is generally less than 0.1 nm/degree, for example 0.06 nm/degree or 0.08 nm/degree. In such systems, 0.01degree temperature stability may be enough to stabilize the laser wavelength. However, in this study, we tried to use the temperature controller to realize the wavelength locking function. So, the laser wavelength would be stabilized at a higher level if we could develop a temperature controller with a higher precision.

Of course, we concluded several studies on the laser temperature controlling based on reviewer’s comment and added sentences to the 4th paragraph of Introduction. Please check the revised manuscript for more details.

  1. In the methodology, the authors should clarify why 3rd harmonic is used. It is well known that even harmonic peak position can indicate the central wavelength position, and have been recently used for high-accuracy gas sensing ([7, 8]). Why they cannot be used to lock the wavelength? If 3rd harmonic is OK, why 1st harmonic cannot be used, for example in ([9]).

Response: Thank you very much for the suggestion. We have added sentences to methodology section to explain why 3rd harmonic is used instead of 2nd or 1st harmonic.

Yes, it is true that even harmonic peak position can indicate the central wavelength position and it can be used to calibrate the laser wavelength in the wavelength scanning mode. In other words, the laser wavelength should be scanned with a bandwidth to get the 2nd harmonic curve to determine the position of extremum point. So, the even harmonic signals are not suitable for dynamic real-time wavelength locking. As a result, odd harmonic signals are always chosen to perform the wavelength locking because the odd harmonic value exhibits monotonicity at the position of absorption line center and such monotonicity is necessary for a PID input. As to the reason why the 3rd harmonic signal is chosen instead of 1st harmonic signal, the 1st harmonic value is very sensitive to the laser intensity which coming from the residual amplitude modulation effect. So, it is not reliable using 1st harmonic signal as a reference to lock the laser wavelength. However, the 3rd harmonic value at the absorption center is almost zero and has nothing to do with laser intensity and gas concentration. It is a constant value which can be used as a reference for laser wavelength locking.

  1. There are existing laser driving ICs, such as AMC7820 from Texas Instruments and WTC3243 from Wavelength Electronics. Why does the authors need to build a circuit to achieve this function?

Response: Thank you very much for the advice. We are more inclined to develop our own temperature controller with higher precision, rather than purchasing a ready-made module. At the same time, we believe that developing a temperature controlling circuit with high precision is highly meaningful. By the way, our lab’s objective is to develop gas sensing instruments by ourselves. So, every part of circuits is designed by ourselves instead of buying commercial modules.

As answered to comment 2, sentences has been added to 4th paragraph to explain this in the revised manuscript.

  1. In Figure 13, the authors should clarify why the SNR peak at 60 dB? Rather than 50 or 70 dB?

Response: We determine the best lock-in bandwidth by the SNR. In the figure 13 (b), the SNR achieved the largest value of 71.2 dB when the lock-in bandwidth is set 60 Hz, which is better than 50 or 70 Hz. (the blue curve)

Supplementary references

[1] A. Farooq, A. B. S. Alquaity, M. Raza et al., “Laser sensors for energy systems and process

industries: Perspectives and directions,” Progress in Energy and Combustion Science, vol. 91,

  1. 100997, , 2022.

[2] C. Liu, and L. Xu, “Laser absorption spectroscopy for combustion diagnosis in reactive flows:

A review,” Applied Spectroscopy Reviews, vol. 54, no. 1, pp. 1-44, 2019.

[3] C. S. Goldenstein, R. M. Spearrin, J. B. Jeffries et al., “Infrared laser-absorption sensing for

combustion gases,” Progress in Energy and Combustion Science, vol. 60, pp. 132-176, 2017.

[4] F. Kehl, V. F. Cretu, and P. A. Willis, “Open-source lab hardware: Driver and temperature

controller for high compliance voltage, fiber-coupled butterfly lasers,” HardwareX, vol. 10,

  1. e00240, 2021.

[5] H. Huang, J. Ni, H. Wang et al., “A novel power stability drive system of semiconductor

Laser Diode for high-precision measurement,” Measurement and Control, vol. 52, no. 5-6,

  1. 462-472, 2019.

[6] L. Xu, C. Liu, W. Jing et al., “Tunable diode laser absorption spectroscopy-based tomography

system for on-line monitoring of two-dimensional distributions of temperature and H2O mole

fraction,” Review of Scientific Instruments, vol. 87, no. 1, pp. 013101, 2016.

[7] Y. Wang, B. Zhou, and C. Liu, “Calibration-free wavelength modulation spectroscopy based

on even-order harmonics,” Optics Express, vol. 29, no. 17, pp. 26618-26633, 2021.

[8] K. Sun, X. Chao, R. Sur et al., “Wavelength modulation diode laser absorption spectroscopy

for high-pressure gas sensing,” Applied Physics B, vol. 110, no. 4, pp. 497-508, 2013.

[9] W. Y. Peng, C. L. Strand, and R. K. Hanson, “Analysis of laser absorption gas sensors

employing scanned-wavelength modulation spectroscopy with 1f-phase detection,” Applied

Physics B, vol. 126, no. 1, pp. 17, 2019.

Reviewer 2 Report

1-      The authors should describe the novel features of their work and how they differ from previous studies.

2-      More details, including the model, must be added regarding the DFB laser system that is being used.

3-      Please explain in the text how you vary the CH4 concentrations.

4-      Can the concentration of CH4 be lowered to the part per billion (ppb) level?

5-      Nothing is mentioned in the text about the uncertainty in the measurements, so please discuss this issue in the text.

The manuscript appears to be clearly and carefully written.

Author Response

Dear Reviewer,

Thank you very much for your efforts on our manuscript. The comments are all valuable and very helpful for revising and improving our study. All the comments are studied carefully and made corrections which we hope to meet with your approval. All the modifications are marked in red in the revised manuscript. The responses to your comments are reported below:

  1. The authors should describe the novel features of their work and how they differ from previous studies.

Response: Thank you very much for the advice. Sentences have been added to the last paragraph of Introduction to explain why we perform this study and how it differs from previous studies. Please check the red-marked words in the revised manuscript.

  1. More details, including the model, must be added regarding the DFB laser system that is being used.

Response: The schematic of experiment system is redrawn and added to the revised manuscript. Please check the new figure 9. In addition, sentences have been added to section 3.2 and 3.3 to provide more details about how the experiment system is conducted. All modifications are marked in red.

  1. Please explain in the text how you vary the CH4 concentrations.

Response: Thank you very much for the comment. Sample CH4 in different concentrations are provided by a gas mixing system used in our lab. The gas mixing system consists of a mixing chamber, two flowmeters, 99.999% pure nitrogen and 1% CH4. We change the flow ratio of two flowmeters to vary the CH4 concentrations. The above information has been supplemented to figure 9 and explained in text in the revised manuscript.

  1. Can the concentration of CH4 be lowered to the part per billion (ppb) level?

Response: In our experiment, we diluted the CH4 to the level of ppm for the linear test with a 1% CH4. However, the accuracy would be affected if the concentration was continuously decreased due to the limited precision of flowmeter. If sub-ppm CH4 sample is needed, we can buy a 0.1% or 0.01% CH4 for diluting. Even though, the ppb level sample gas is very difficult to generate considering the molecular adsorption effect. In this study, only linear test and measurement uncertainty are discussed to compare the standard WMS and wavelength locked WMS. Trace CH4 in ppb level is not necessary.

  1. Nothing is mentioned in the text about the uncertainty in the measurements, so please discuss this issue in the text

Response: Thank you very much for the suggestion. We have added the discussion on uncertainty to the section 3.3 of revised manuscript and marked in red.

Reviewer 3 Report

The following major comments need to be addressed.

1. The abstract is more like a conclusion. The abstract should be a summary of introduction, problem statement, methodology, results and conclusion. In short, it should be an executive summary.

2.  At the end of the introduction, there is no problem statement, and contributions.

3. In the entire manuscript, there are no details of PID controller. However, the authors have mentioned PID is the controller used and its performance.

4. The controller is also missing in Figure 4.

5. Provide more details on hardware specifications.

6. How are the parameters in Table 1 chosen?

7. In Figure 8, how is a sharp change in temperature possible in practical?

8. There is no comparison with literature.

9. No numerical analysis

10. Add any drawbacks of existing and propose future work

Requires a moderate editing of English language 

Author Response

Dear Reviewer,

Thank you very much for your efforts on our manuscript. The comments are all valuable and very helpful for revising and improving our study. All the comments are studied carefully and made corrections which we hope to meet with your approval. All the modifications are marked in red in the revised manuscript. The responses to your comments are reported below:

  1. The abstract is more like a conclusion. The abstract should be a summary of introduction, problem statement, methodology, results and conclusion. In short, it should be an executive summary.

Response: Thank you very much for the advice. We have polished the abstract follow your suggestion.

  1. At the end of the introduction, there is no problem statement, and contributions.

Response: Problem statement and contribution have been added to the last paragraph of the Introduction section and marked in red.

  1. In the entire manuscript, there are no details of PID controller. However, the authors have mentioned PID is the controller used and its performance.

Response: Thank you very much for the comment. The PID controller is a very important of temperature controller. Figure 4 is modified to reflect the PID part and sentences are added to section 3.1 to explain how the PID controller works.

  1. The controller is also missing in Figure 4.

Response: As answered to comment 3, the PID controller is indicated in the new figure 4 of revised manuscript.

  1. Provide more details on hardware specifications.

Response: More details about hardware and experiment system have been supplemented to the section 3.1, 3.2 and 3.3 in the revised manuscript.

  1. How are the parameters in Table 1 chosen?

Response: R1 and R2 are set 10  to match the thermistor Rt to constitute the balanced H-bridge. Rset is a high-precision rheostat with low temperature drift (50 ppm/degree) to guarantee the precision of Vset.  are key parameters of PID controller, which are chosen based on our previous experience reported in IEEE Sensors Journal, doi: 10.1109/JSEN.2023.3246508.  constitute the basic LC filter. The above specifications are supplemented to the revised manuscript.

  1. In Figure 8, how is a sharp change in temperature possible in practical?

Response: Thank you very much for the comment. Such sharp change of laser temperature does not happen in practical application. We put figure 8 here to display the speed of our developed temperature controller. It takes only 9 seconds for the temperature controller to stabilize the DFB laser from ~36  to ~30 . In practice, the wavelength drift of DFB laser is gradual and tiny when it operates in free-running mode. Therefore, minor adjustment of laser temperature is able to calibrate such wavelength drift. In another word, the response speed revealed in Fig. 8 is high enough to realize efficient wavelength locking in this study. The above demonstration was added to the last paragraph of section 3.1.

  1. There is no comparison with literature.

Response: comparison analysis has been added to the revised manuscript in the last paragraph of experiment. The recently published reference [30-32] are provided for comparison.

  1. No numerical analysis

Response: numerical analysis was performed on the experimental results, about the temperature controlling stability, wavelength locking precision, CH4 detection accuracy and signal to noise ratio.

  1. Add any drawbacks of existing and propose future work

Response: Limitations and countermeasures have been analyzed in the second paragraph of Conclusion.

Round 2

Reviewer 1 Report

The authors have addressed the previous comments. The paper can be accepted for publication.

Author Response

Dear Reviewer,

Thank you very much for your expertise again. The final manuscript was minor revised to get rid of grammar and spelling mistakes. 

Yours sincerely,

Dr. Fupeng Wang

Reviewer 2 Report

In response to my previous suggestions and concerns, the authors have made reasonable changes to the manuscript. Overall, the manuscript reads well and clarifies the authors' work. In my opinion, the manuscript contains currently all information and is ready for publishing in the Journal " Sensors " as a regular article. 

Overall, the manuscript reads well 

Author Response

Dear Reviewer,

Thank you very much for your expertise and effort on our study again. The manuscript was minor revised to get rid of grammar and spelling mistakes. We believe this version is a final one without any error.

Yours sincerely,

Dr. Fupeng Wang

Reviewer 3 Report

All my comments have been addressed. However, the resolution of figures has not been improved. 

Minor English editing is required. 

Author Response

Dear Reviewer,

Thank you very much for your expertise and comments on our paper again. The manuscript was minor revised to get rid of grammar and spelling mistakes. All the figures used in this paper are uploaded separately. We believe this version is a final one without any error.

Yours sincerely,

Dr. Fupeng Wang